# HLGM: A Novel Methodology For Improving Model Accuracy Using Saliency-Guided High and Low Gradient Masking

1st Ali Karkehabadi
*ECE Department*
*UC Davis*
USA
akarkehabadi@ucdavis.edu

2nd Banafsheh Saber Latibari
*ECE Department*
*UC Davis*
USA
bsaberlatibari@ucdavis.edu

3rd Houman HomayouN
*ECE Department*
*UC Davis*
USA
hhomayoun@ucdavis.edu

4th Avesta Sasan
*ECE Department*
*UC Davis*
USA
asasan@ucdavis.edu

*Abstract*—In this paper, we introduce the High and Low Gradient Masking (HLGM) approach, a groundbreaking saliency-guided training method that effectively enhances both the accuracy and the quality of saliency maps in computer vision models. This method stands apart from traditional saliency-guided training, which often compromises accuracy. HLGM employs a novel two-phase process: initially, it involves regular training without gradient masking, followed by an accuracy boosting phase. This phase alternates between masking high gradient information to encourage diverse learning pathways, and masking low gradient information to reduce background noise and strengthen crucial synoptical pathways. The effectiveness of HLGM is validated through a unique metric that measures the alignment of high-fidelity saliency feature maps with labeled objects in images. Our comparative analysis against baseline models and current advanced techniques demonstrates substantial improvements in both model accuracy and saliency mapping. HLGM not only outperforms conventional training methods in accuracy but also advances model interpretability, positioning it as a pivotal tool in the pursuit of explainable AI in machine learning.

*Index Terms*—Deep Learning, Interoperability, Saliency-Guided Training

## I. INTRODUCTION

Deep learning's transformative effect on society is attributed to its ability to discern complex patterns from vast datasets [28]. The adoption of Deep Neural Networks (DNNs) has led to marked improvements in prediction accuracy and decision-making, as proved by empirical findings. This progress has accelerated both social and technological advancements. However, due to DNNs' black-box nature, there are concerns about their reliability. This has stimulated significant academic interest in understanding their behavior and the criteria they rely on for generating outputs. Being able to give clear reasons is really important in fields like medicine, neurology, finance, and autonomous driving [32]. These explanations not only heighten the understanding and trust in models but also assist in their debugging and refinement [32]. As a result, a substantial body of research has been dedicated to advancing interpretability techniques for DNNs [4], [8], [11], [13]. One common approach involves finding influential input

features that play a significant role in classification outcomes, utilizing methods like saliency maps that often employ gradient calculations [6], [11], [13], [25]. These maps assign importance values to features, highlighting their influence on predictions. The presence of noise or distracting elements can compromise the clarity of saliency maps, leading to diminished clarity and precision. Singh et al. [4] introduced methods that use higher-order backward gradients, enhancing saliency map insights. Kindermans et al. [5] enhanced their findings through multiple gradient computations. Notably, the SmoothGrad technique addresses saliency noise by repeatedly adding noise to inputs and then averaging the produced saliency maps [11]. Other methods, including integrated gradients [13], DeepLIFT [25], and Layer-wise Relevance Propagation [8], modify backpropagation with alternate gradient functions [3]. However, the effectiveness of these techniques is tied to their reliability [5]. Saliency maps lose trustworthiness if they show large shifts from minor input or model changes, as emphasized by Ghorbani et al. [22]. Thus, comprehensive validation is essential when developing new interpretability methods to ensure their dependability [15]. The success of these methods can vary based on data types and model architectures, requiring continual method refinement [24]. Enhanced interpretability extends beyond understanding individual predictions; it aims to interpret a model's overall decision-making logic [9]. For instance, to achieve this, some researchers are exploring distillation techniques to transform intricate neural networks into interpretable models like soft decision trees [1]. Saliency itself does not directly influence the accuracy of a model. However, if a saliency map indicates that the model is concentrating on irrelevant parts of the data, modifications can be implemented to potentially enhance its accuracy. Saliency Guided Training (SGT) seeks to achieve dual objectives: ensuring the model displays significant saliency while upholding an impressive level of accuracy. However, adopting saliency guided training could result in a notable reduction in accuracy [12]. In this study, we present a novel training approach denoted as High and Low Gradient Masking (HLGM) masking, a variant of saliency guided

masking that improves the model accuracy while generating equal or better quality saliency map solution compared to prior art.

## II. RELATED WORKS

Interpretability in machine learning seeks to make model decisions understandable to humans, which is crucial for ensuring that people can comprehend why models make certain decisions. Deep learning models, while highly accurate, often lack transparency, hindering our ability to justify their predictions logically. Interpretability aims to address this by enhancing model transparency and functionality, enabling us to bridge the gap between complex model processes and human decision-making, thus ensuring the reliability of AI systems [26]. [10] introduced "grafting," an approach to feature selection that seamlessly integrates into gradient descent training. This method employs an incremental, gradient-based strategy, iteratively selecting and adding features while optimizing predictive models. Grafting is efficient with data points and features, compatible with both linear and non-linear models for classification and regression. The techniques used in grafting complement approaches like [1]'s stochastic gradient descent-based soft decision tree, to [27]'s human-annotated "rationales" for model justification. Other notable methods include saliency learning [2], input gradient modulation [9], class differentiation in CNN training [16], cutout regularization and additive feature significance approach [6]. [2] set out to train the model to make accurate predictions based on sound reasoning. They achieved this by incorporating explanation training and ensuring that the model's explanations aligned with the true, valid explanations. Their primary objective was to guide the model in directing its attention to relevant information, thereby preventing it from being swayed by irrelevant statistical biases in the data. Their research predominantly emphasized positive explanations, meaning that they aimed for explanations that highlighted information contributing positively to the label prediction. In recent years, tools like LIME have emerged from efforts to explain black-box models, shedding light on the implicit rules guiding predictions. Such tools were valuable in discerning situations where models made correct predictions for the wrong underlying reasons. However, limitations became evident as these methods struggled to scale to explain entire datasets and could not rectify the issues they uncovered. To address these challenges, [9] introduced a novel approach. Their method efficiently explained and regularized differentiable models by examining and selectively penalizing input gradients. These penalties were applied in two ways: through expert annotation and an unsupervised approach. The result was the creation of multiple classifiers with distinct decision boundaries, adding depth and nuance to model interpretation. Researchers have delved into the realm of gradient-based attention modeling, where attention maps have emerged as a powerful tool for deciphering convolutional neural networks. While these methods have been successful in effectively pinpointing specific classes of interest, they encountered a significant problem: attention maps tended to overlap substantially across different classes, leading to visual confusion. To tackle this issue, [16] introduced a fresh framework in their paper, making class-discriminative attention an integral part of the learning process. Their key advancements included the introduction of novel learning objectives aimed at improving attention distinctiveness and consistency across network layers. These innovations ultimately resulted in enhanced attention separability and a reduction in visual ambiguity. [6] introduced a simple framework aimed at generating local explanation vectors suitable for various classification methods. The main objective was to aid in understanding the prediction outcomes for individual data points. These local explanations were crafted to reveal the pertinent features influencing predictions at specific locations within the data space. Furthermore, this approach had the ability to uncover local idiosyncrasies that were frequently disregarded when examining the global perspective, often due to the influence of offsetting factors.

However, challenges persist. While gradient techniques exhibit many desirable traits, their application to visual models often results in the generation of misleading or noisy pixel attributions in regions that have no relevance to the predicted class [23]. Such disparities in saliency maps might be attributed to minor local derivative fluctuations [13]. [37] elucidated the contrast between low-level features, which can capture superfluous elements, and high-level features, which prioritize semantic information. To emphasize essential data, stringent criteria are necessary when extracting saliency maps using low-level features. Integrating saliency ensures that models emphasize pivotal data, potentially boosting performance and fostering generalization. [34] posit that verbalizing saliency maps enhance human comprehension, providing more digestible feature attribution explanations than traditional methods. [35] noted remarkable advancements in the generalization ability of Soft Actor-Critic agents, surpassing contemporary benchmarks in training efficiency, generalization gap, and policy interpretability. [12] proposed an innovative approach called saliency-guided training. In their work, they introduced a novel algorithm that integrates interpretability into the training process. This algorithm aims to improve the accuracy of models while also considering the concept of saliency.

**Algorithm 1** delineates the SGT process, which harnesses saliency data to bolster the training of a neural network model represented as $f_\theta$. The model's output, given $X$, is denoted as $f_{\theta_i}(X)$ whose parameters $\theta$ are updated based on the gradient of the loss. Within this framework, the Kullback-Leibler divergence, $\mathcal{D}_{\mathcal{KL}}(p|q)$, serves as a metric to discern variations between two probability distributions, specifically contrasting the original output distribution $f_\theta(X)$ with its modified counterpart $f_\theta(\widetilde{X})$. The masking function $M_k(I, X)$ is instrumental, omitting the least impactful $k$ features from the input $X$ based on the rank $I$, which denotes feature significance as per gradients. This yields the modified input $\widetilde{X}$, having the less pertinent features excluded via the mask $M_k(I, X)$. The holistic loss function $L_i$ used in training amalgamates two elements: the conventional loss metric $\mathcal{L}(f_\theta(X), y)$, assessing

**Algorithm 1** Saliency Guided Training [12]

**Inputs:**
- ⊸ Training samples $X$,
- ⊸ Number Masked Features $k$,
- ⊸ Learning Rate $\tau$,
- ⊸ KL-Divergence weight $\lambda$,
- ⊸ Number of epochs $N$
- ⊸ Initialize $f_\theta$
- ⊸ Preload or randomize for new training

**for** $i = 1$ *to* $N$ **do**
  **for** *minibatch* **do**
    Extracting sorted index $I$ of the gradient of output w.r.t the input.
    $I = \text{sort}(\nabla_X f_{\theta_i}(X))$
    Masking bottom $k$ features of the original input.
    $\widetilde{X} = M_k(I, X)$
    Computing the loss function.
    $L_i = \mathcal{L}(f_{\theta_i}(X), y) + \lambda \mathcal{D}_{\mathcal{KL}}(f_{\theta_i}(X) \| f_{\theta_i}(\widetilde{X}))$
    Using the gradient to update network parameters
    $f_{\theta_{i+1}} = f_{\theta_i} - \tau \nabla_{\theta_i} L_i$
  **end**
**end**

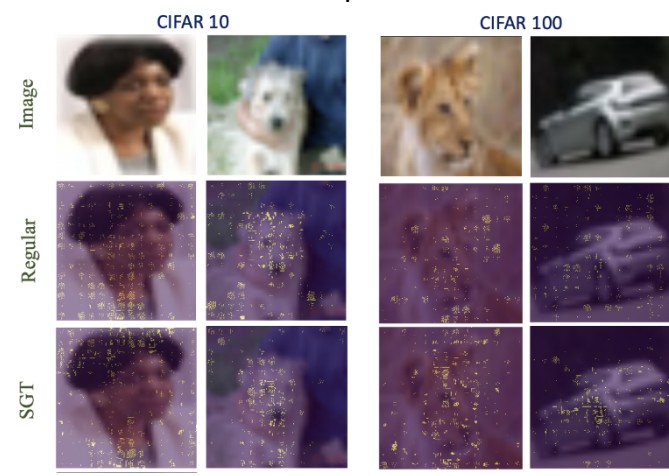

Fig. 1: This image displays the saliency map produced by the SGT method [12], a prior art solution, compared with a regular model. Although the SGT has better performance, both models show that many of the top gradients do not correspond to the main object, and many pixels of the main object are not considered.

the model's efficacy on the pristine input $X$ against labels $y$, and a regularization component rooted in the Kullback-Leibler divergence, advocating for congruence between the output profiles of $X$ and $\widetilde{X}$.

## III. METHODOLOGY

Fig. 1 shows the application of SGT (Algorithm 1) to various images. The saliency map is superimposed on each blurred image, highlighting input features with the highest gradient, indicating their significance for the model's prediction. As illustrated in this saliency map, the object features deemed significant to the learning model (as highlighted by the saliency map) are only a subset of the features by which the object could be classified.

This means the object is identified using only a subset of its features. The first question is whether we can use the saliency information to push the model to learn new features or be able to identify the object in many distinct ways. The second observation is that the saliency map contains many points outside the object that are not logically related to the identified object. This is another area for improvement to understand if we can lower the model's excitation to non-contributing features.

### A. Regularization and Denoising by HLGM

In this work, we propose using a saliency map as a regularization parameter. The regularization approach we propose pushes the model to identify objects in many different ways. We propose achieving this by a form of data augmentation that removes the high gradient input features (pixels and surrounding area) from the input image and continues the training process to explore a different way to learn the model. By identifying new features, the model can rely on a larger set of features for identifying objects and improve its robustness. As a second step, we propose using the saliency map as a denoising mechanism and removing input features with a low gradient from the input image. This will produce a new set of input images. By using denoised images, the model is boosted as it enforces the excitations of the synaptic sub-network responsible for the classification of the target class. We propose performing the regularized SGT and denoised SGT sequentially, allowing the model to learn new features and subsequently boost network performance by enforcing newly learned weights. More specifically, we propose starting with a conventional training process, without regularized or denoised SGT, and moving to the proposed iterative regularized-denoised SGT iteration when the model performance no longer improves. Algorithm 2 captures the details of our proposed regularized and denoising saliency guided training solution. Details of this algorithm are described in the following section. In summary, we force the model to learn additional important pixels by initially masking high-gradient pixels (regularized SGT). After a few epochs, when the model has learned these new important pixels, we then mask low-gradient pixels (denoised SGT), enabling the model to learn both the new and previously identified important pixels.

### B. Detailed Explanation of Proposed Approach

Algorithm 2 takes as input the training database, the input mode (pre-trained or randomized weights), the learning rates, and several hyperparameters including k that indicates how many features will be masked during regularized training, $\gamma$ that indicates the weight KL-divergence in the loss function. The training process in this algorithm has two major steps: conventional training and Regularize-and-Denoising SGT. As

illustrated in the equations below, the conventional training step uses cross-entropy loss to update the weights iteratively until the loss is flat.

After the decrease in the model loss reaches its plateau through conventional training (checked by monitoring the rate of change in loss), our proposed solution enters its second training phase. The SGT training step in our solution is composed of two stages that are iteratively and sequentially executed: "Regularized SGT" and "Denoising SGT". To start the SGT step, as highlighted in the Alg.2, we first compute the input gradients and then sort the inputs based on gradients. In the following equation, $I$ is the sorted input gradient. After sorting the gradients, we enter either the regularized or denoising SGT. Implementation of regularized denoising is simply done by masking the $k$ top pixels in the sorted list (those with the highest gradient) and the implementation of denoising SGT is done by masking the bottom $k$ input features (those with the least gradient).

In Algorithm 2, masking the top $k$ high gradient pixels is done by the function $M_t(k_U)(\cdot)$, and masking the bottom $k$ pixels is done using the function $M_b(k_B)(\cdot)$. Different values of $k$ can be used in each epoch. As the model learns better features over time, the number of top features masked should be reduced. This is achieved by using the parameter $\alpha$ to decrease the number of top features masked. When the masking function is applied, it generates a new input image denoted by $\widetilde{X}$. Similar to the original SGT, the $\widetilde{X}$ is then used to compute the Kullback-Leibler (KL) divergence.

In Algorithm 2, we utilize several equations essential for the explanation of the model's operation. Equation 1 defines the Kullback-Leibler (KL) divergence, which measures how one probability distribution $P$ diverges from a second expected distribution $Q$. This metric is crucial for comparing the neural network's outputs on original versus perturbed inputs, helping to refine the model by minimizing information loss:

$$D_{KL}(P\|Q) = \sum_{x \in X} P(x) \log\left(\frac{P(x)}{Q(x)}\right) \quad (1)$$

Equation 2 addresses the sorting of gradients derived from the network's predictions to rank the importance of input features. This ranking guides the selective masking of less influential features, thus focusing the model's learning on crucial inputs:

$$I = \text{sort}(\nabla_X f_{\theta_i}(X)) \quad (2)$$

For the loss of the model, we combine Cross-Entropy Loss and Loss with KL Divergence. The classification loss minimizes the KL divergence between $f_\theta(X)$ and $f_\theta(\widetilde{X})$ to ensure that the trained model produces similar output probability distributions over labels for both masked and unmasked inputs. The optimization problem is formulated as follows:

$$\sum_{i=1}^{n}\left[ L(f_\theta(X_i), y_i) + \lambda D_{KL}(f_\theta(X_i)\|f_\theta(\widetilde{X}_i)) \right] \quad (3)$$

---

**Algorithm 2** HLGM: High and Low Gradient Masking

**Inputs:**
- ⊸ Number Masked (bottom and top) Features $k_b, k_t$
- ⊸ Number of Reduced Masked Top Features $\alpha$
- ⊸ Learning Rate $\tau$
- ⊸ KL-Divergence weight $\lambda$
- ⊸ Episode-Size $S_e$
- ⊸ Number of epochs $N$
- ⊸ Training samples $X$
- ⊸ Initial count of denoising epocks withing each SGT episode $de$
- ⊸ Initialize $f_\theta$
- ⊸Preload or randomize for new training
- ⊸ Phase $\leftarrow Training$

**for** *(epoch $i = 1$ to N)* **do**
  **if** *(Phase==Training)* **then**
    **in Traditional State**
    Computing the loss function.
    $L_i = \mathcal{L}(f_{\theta_i}(X), y)$
    Using the gradient to update network parameters.
    $f_{\theta_{i+1}} = f_{\theta_i} - \tau\nabla_{\theta_i}L_i$
    **if** *(loss does not improve in last 20 epochs)* **then**
      $Phase \leftarrow$ SGT
      $episode = 0$
    **end**
  **else**
    **in SGT State**
    episode ++
    Compute and sort the gradients into vector $I$
    $I = \text{sort}(\nabla_X f_{\theta_i}(X))$
    **if** *(episode $< S_e - de$)* **then**
      Regularization: Masking top $k$ features.
      $k_t = k_t - \alpha$
      $\widetilde{X} = M_{K_t U}(I, X)$
    **else**
      Denoising: Masking bottom $k$ features.
      $\widetilde{X} = M_{K_b B}(I, X)$
      Computing the loss function.
      $L_i = \mathcal{L}(f_{\theta_i}(X), y) + \lambda\mathcal{D}_{\mathcal{KL}}(f_{\theta_i}(X)\|f_{\theta_i}(\widetilde{X}))$
      Using the gradient to update network parameters.
      $f_{\theta_{i+1}} = f_{\theta_i} - \tau\nabla_{\theta_i}L_i$
      increase denoising epochs in the next episode
      **if** *(($episode == S_e$) & ($de \leq S_e$)* **then**
        $de$ ++
        start a new episode
        $episode = 0$
      **end**
    **end**
  **end**
**end**

---

The regularized and denoising SGT are executed episodically with each episode containing $S_e$ epochs. During the first episode, $S_e - de$ epochs are regularized-SGT, and the $de$ epoch is denoising. In each subsequent episode, the number of regularized epochs reduces by one and the denoising epoch is added by 1, keeping the episodes of $S_e$ epoch but gradually

shifting the focus from regularization to denoising, until in the last few episodes we only run denoising episodes. In our results section, we considered episodes of 5 epochs to generate the results.

## IV. EXPERIMENTS

In this section, we initially provide a concise overview of the datasets employed to evaluate our research. Subsequently, we delve into the specific learning model structures utilized for this evaluation. We further explore the effectiveness of our proposed approach in enhancing model accuracy across various datasets and learning models. Lastly, we contrast the saliency map produced by our method with the original SGT as referenced in [12].

### A. Datasets for Evaluation

**CIFAR-10:** The CIFAR-10 dataset [28] includes 60k low-resolution RGB images, categorized into 10 classes. We employed 50k images to train and 10k to test.

**CIFAR-100:** The CIFAR-100 dataset [28] consists of 60k low-resolution RGB images categorized into 100 classes, grouped into 20 superclasses. We split it into 50k training and 10k test images.

**Oxford-IIIT:** The Oxford-IIIT Pet Dataset [7] contains 7,349 images of cats and dogs, with 37 breed categories. We used this dataset to assess our model's performance in pet breed classification.

### B. Machine Learning Models Used in This Work

**Shallow CNN (2-Layers):** This learning model is used in the original SGT paper [12]. Using this model allows us to compare the performance of our solution against the original SGT on the same model and dataset. The shallow CNN model is a simple Convolutional Neural Network (CNN) architecture, consisting of two CNN layers with 3x3 kernels and a stride of 1, followed by 2 Fully-Connected (FC) layers. The model also includes two dropout layers with dropout rates set at 0.25 and 0.5, respectively.

**Deep CNN (ResNet18):** Use of ResNet18 [29] allow us to evaluate our solution on deep CNNs. In this work, we used a pre-trained ResNet18 on the ImageNet dataset [21]. Subsequently, we integrated a 100-neuron output classifier for the Cifar100 and a 37-neuron output classifier for the Oxford-IIIT datasets in the model.

**Tiny Transformer:** Transformers, a type of deep learning architecture, have transformed natural language processing with self-attention mechanisms [38]. Their capacity to grasp long-range dependencies has made them pivotal in achieving top-notch results in various machine learning applications. Models like BERT [18], GPT-3 [14], and GPT-4 [17] have significantly advanced NLP, inspiring their application in computer vision. Attention-based models for image recognition [31] and object detection [39] have outperformed prior CNN models. To assess our approach for training attention-based models, we experimented with the compact Tiny Transformer model [30] (configured as $(L = 12, d = 192, h = 3)$), known

| Model | LR | Batch Size | Opt |
|---|---|---|---|
| Shallow CNN | 0.1 | 256 | Adadelta |
| ResNet-18 | 0.01 | 256 | SGD |
| DeiT-tiny | 0.001 | 256 | Adadelta |

TABLE I: Summary of DNN architectures and training parameters

for its efficiency and comparable performance to deep CNNs. We adopted this 'tiny' configuration from the pre-trained 'deit' model [1], initially trained on ImageNet [29]. For our datasets, we replaced the classification head with a 10-neuron classifier for CIFAR-10 and a 100-neuron classifier for CIFAR-100.

### C. Training Details

For all architectures, the training was done on an NVIDIA A100 GPU. Each model was trained for 100 epochs with a batch size of 256 samples. In our training, we used the Adadelta [33] optimization algorithm with a learning rate of 0.1. To enhance gradients, low gradient pixels were substituted with random values within the remaining pixel spectrum during each epoch, with gradient values computed using the Captum library [36]. For the Tiny Transformer, the same training details were maintained, except that the optimizer used as Stochastic Gradient Descent(SGD) [20] with a learning rate of 0.001. Similar to Shallow CNN (2-Layers), gradient enhancement involved substituting low gradient pixels with random values using the Captum library [36]. Table I presents a summary of the training specifics for the examined models. For the number of masked bottom features, we use 50% of the input features:

$$M_b(k_B)(\cdot) = 50\% \text{ of input features}$$

For the number of masked top features, we use 20% of the input features:

$$M_t(k_U)(\cdot) = 20\% \text{ of input features}$$

and apply a reduction of 2%, resulting in the number of masked top features being reduced by 2% ($\alpha = 2\%$)

Fig 2 displays the accuracy progression for the Shallow CNN model on the CIFAR10 dataset. As illustrated in this image and expected from Alg 2, the training goes into 3 phases. The first phase, highlighted in light grey, is the traditional training. The training stops when the test accuracy does not improve for the past 20 epochs. The Algorithm then enters the SGT-based training, consisting of a sequence of regularized and denoising stages. The regularized-SGT, highlighted in purple, removes the high saliency pixels from the input image, feeding the model with a modified image and forcing the model to learn to classify the output using a different set of features. The purpose of the regularized-SGT is to improve the robustness of the model and force the model to learn many different features by which it can classify an input image. The regularized-SGT is followed by a denoising-SGT. The denoising-SGT phase is highlighted in light yellow. In denoising-SGT the low gradient input pixels are masked,

---

[1]https://github.com/facebookresearch/deit

allowing the model to update and improve the weights on the synaptic subnetwork responsible for image classification and also improve its saliency map. The denoising-SGT is similar to the original SGT paper in [12] and it is the addition of regularized-SGT and sequential invocation of regularized and denoising SGT that makes our work different. The episodic application of regularized and denoising SGT continues and in each episode, the number of regularized-SGT epochs is reduced by 1. This allows the model to explore and find many different ways in the first episode to classify the input, gradually reduce the exploration of new features, and start improving the saliency map in the last episodes. In this image, we have shown both the training and test accuracies. As illustrated there is a visible deep in the training accuracy when we start the regularized-SGT as a model can no longer use the features it previously learned for classification of input. However, as illustrated, the model start to recover quickly and learn new features. Starting from the first purpose zone (regularized-SGT), we show two test accuracies. The red line is the test accuracy with SGT-based training, and the black is the continuation of the traditional training. Through this illustration, we show the improvement in test accuracy as a result of using our proposed HLGM algorithm. In this figure, and all subsequent figures, there are two comparisons. The first comparison is the comparison of the max test accuracy between our solution and conventional regardless of which epoch it takes place. The second comparison is the accuracy comparison between the two at the end of training (100 epochs). As illustrated in this figure, and in all subsequent figures, you will see that the HLGM results in improving the accuracy while simultaneously improving the saliency map. The observed improvement in the accuracy, depending on the learning model and data set changes and in our results varies between 1 to 4.5% improvement in accuracy which is considered quite substantial.

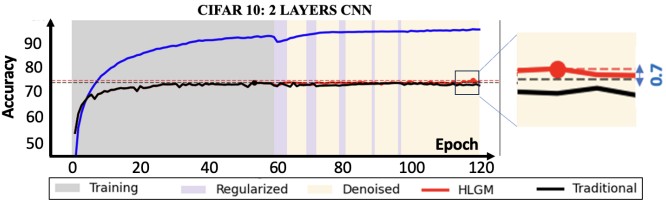

Fig. 2: Comparison of Training and Testing Accuracy on the CIFAR-10 Dataset Using CNN Model with HLGM and SGT Algorithms: After a certain number of epochs, the accuracy stabilizes, as indicated by the black area. Initially, the HLGM algorithm's masking of high gradients leads to misclassification, reducing accuracy because the model focuses on other significant pixels, as shown in the blue area. As training progresses, the model shifts its focus to low-gradient areas, identifying and learning from key pixels, represented by the yellow area. This shift occurs because the model is attempting to identify more important pixels instead of those in the black area.

Figure 3 captures the impact of HLGM training on the transformer model on CIFAR-10 and CIFAR-100 datasets. As illustrated in this figure, HLGM causes a significant improvement in test accuracy compared to conventional training in both datasets.

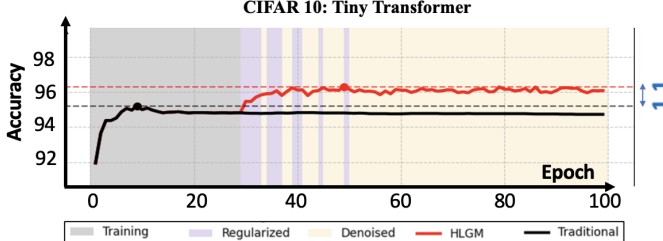

Fig. 3: Comparison of Training and Testing Accuracy on CIFAR-10 Using CNN with HLGM and SGT Algorithms: Accuracy doesn't increase after several epochs, as indicated by the black area. Initially, HLGM's high gradient masking (blue area) leads to increased accuracy. Subsequently, the model adapts by focusing on other relevant pixels. Finally, by targeting low gradient regions (yellow area), it effectively learns from key pixels.

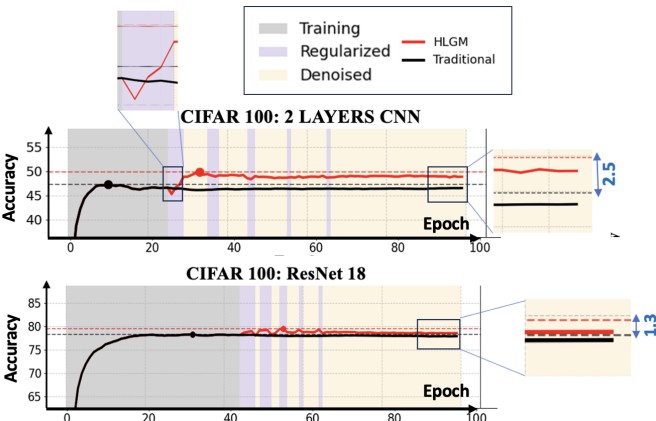

Fig. 4: In CIFAR-100: Accuracy peaks after several epochs, highlighted by the black. At first, the HLGM algorithm's actions led to mistakes and decreased accuracy, but by learning other important pixels accuracy rises (indicated in the blue area) before the model focuses on low gradient areas (highlighted in the yellow area).

Figure 4, captures the result of using HLGM training with shallow CNN (2-layer) CNN and Deep CNN (ResNet-18) on the CIFAR-100 dataset. Our solution causes a notable enhancement in test accuracy. Figure 5 captures the HLGM on the Tiny Transformer model using the CIFAR-100 dataset. To use the pre-trained tiny transformer, we resized the input images from 32x32 to 224x224 pixels and used them as input for both traditional and HLGM training. As illustrated the test accuracy of HLGM training surpasses that of traditional confirming the applicability of this accuracy and saliency enhancing solution on transformers. Table II provides a comparative analysis, indicating the superior performance of our proposed HLGM solution across all models and datasets.

## V. SALIENCY COMPARISON

HLGM training not only enhances model accuracy but also advances our understanding of the model's functionality by providing improved saliency maps. These maps are crucial in elucidating how the model processes input images. To generate saliency maps for a given image, we adopted the approach detailed in [12]. This visualization technique highlights the

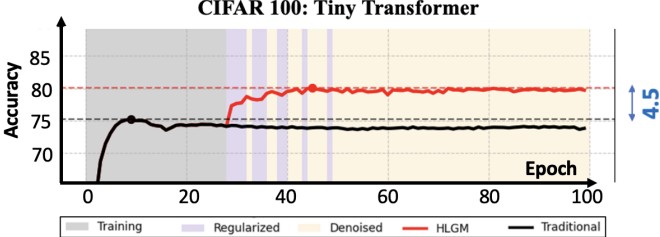

Fig. 5: CIFAR-100 by DeiT-tiny: Accuracy reaches its peak after several epochs, as illustrated by the black line. By using high-gradient masking, due to the numerous pixels and the significant size of 224x224, accuracy subsequently increases, as depicted in the blue area. Subsequently, the model emphasizes low-gradient regions (marked in the yellow area).

| Model | Dataset | Baseline | SGT | HLGM |
|-------|---------|----------|-----|------|
| 2 Level Conv | CIFAR10 | 74.13% | 73.34% | **74.72**% |
| 2 Level Conv | CIFAR100 | 46.51% | 45.36% | **49.87**% |
| ResNet-18 | CIFAR100 | 78.30% | 78.92% | **79.58**% |
| DeiT-tiny | CIFAR 10 | 94.72% | 95.98% | **96.44**% |
| DeiT-tiny | CIFAR 100 | 75.16% | 78.86% | **80.10**% |

TABLE II: Comparison of Models Accuracy when using Baseline(Regular), SGT, and HLGM approach for model training on CIFAR-10 and CIFAR-100 Datasets.

high-gradient points within the input image—areas deemed more significant—thereby offering a visual interpretation of the features the model focuses on during inference. Figure 6 presents a comparative analysis of this visualization across selected images from the CIFAR-10 and CIFAR-100 datasets. It demonstrates the effects of standard training, SGT, and HLGM on saliency map creation.

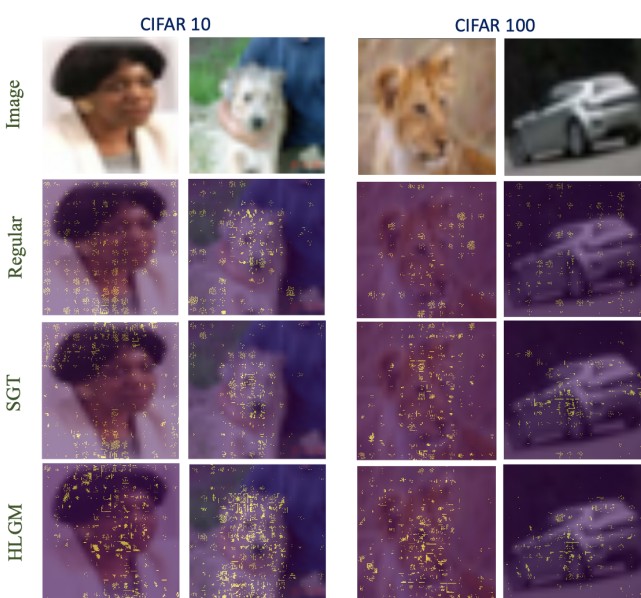

Fig. 6: Performance Comparison of Saliency: HLGM Model, SGT Model, and Regular Model on CIFAR-10 and CIFAR-100 Datasets. High gradient points are highlighted in yellow, revealing that the HLGM method exhibits higher gradients on the main object compared to the regular and SGT methods.

Visually, saliency maps generated by HLGM demonstrate a more distinct correlation with image pixels that are relevant to label information. This observation offers a qualitative per-

spective; however, to provide empirical support for our claim that HLGM yields enhanced saliency maps, we incorporate a quantitative analysis. To this end, we introduce a novel metric called Saliency In the Box (SIB). SIB is particularly applicable to datasets containing object bounding box information. Essentially, the SIB(x) metric quantifies the proportion of high-gradient points residing within the bounding box. Here, 'x' represents a thresholding mechanism.

To calculate SIB, we begin by computing the saliency values for each pixel. Subsequently, we rank these pixels according to their saliency values. Let's denote H(x) as the highest x% of these pixels, ranked by gradient value. Assume BB represents the bounding box of the object in the input image. Then, SIB is the percentage of points in H(x) that are located within BB. Formally, SIB can be defined as follows:

$$\text{SIB }(x) = \frac{\text{Number of points in H(x) inside BB}}{\text{Number of points in H(x)}} \qquad (4)$$

Figure 7 depicts the model's emphasis on the primary object in Oxford-IIIT Dataset [7], using the distribution of top 5% highest gradients, SIB(5), within and outside the bounding box.

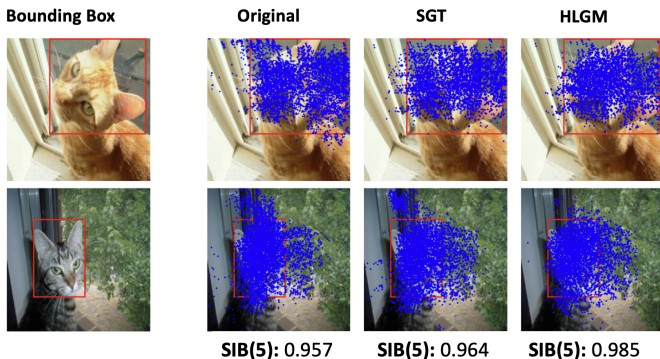

SIB(5): 0.957    SIB(5): 0.964    SIB(5): 0.985

Fig. 7: Distribution of high-gradient points within and outside the object's bounding box demonstrates the model's emphasis on the primary object in the images in the Oxford-IIIT Dataset [7]. This outcome is based on the top 5% highest gradients in the image, and the percentage results indicate the proportion of these high gradients located within the bounding box.

Figure 8 showcases the calculated SIB(1) and SIB(5) for the Oxford-IIIT dataset [7], utilizing the ResNet architecture [29] as described in Section IV-B. We trained ResNet with three different approaches: standard (baseline), SGT, and HLGM. Each model, post-training, was employed to generate saliency maps for all images in the test set, from which we computed SIB(1) and SIB(5) metrics. On the left side of the figure, we present box plots for SIB(1) and SIB(5). As anticipated, the HLGM method exhibits a higher mean in the SIB box plot compared to both SGT and baseline training. This suggests that the saliency maps generated by HLGM more frequently encompass the object within the bounding box, indicating focused attention of the model on the object rather than the background during inference. This also implies that classification is predominantly based on the object's primary features rather than environmental artifacts, marking a step towards enhanced explainability.

| Model | Baseline | SGT | HLGM |
|---|---|---|---|
| Median(SIB) for top 0.5% | 0.58 | 0.62 | **0.65** |
| Median(SIB) for top 1% | 0.55 | 0.57 | **0.59** |
| Median(SIB) for top 2% | 0.49 | 0.53 | **0.56** |
| Median(SIB) for top 5% | 0.45 | 0.47 | **0.50** |
| Median(SIB) for top 10% | 0.41 | 0.43 | **0.46** |

TABLE III: Median SIB value for Baseline( Regular), SGT, and HLGM training for a range of SIB values.

The right side of the figure offers an alternative representation of this data. Here, we computed the SIB value, and then for 120 randomly selected images from the test set, we sorted them based on the SIB values derived from the baseline model. In this visual, each point on the x-axis represents an image, and the three circles on the y-axis associated with each x-axis point correspond to the SIB values for baseline, SGT, and HLGM models, respectively. This clearly illustrates that, in the vast majority of cases, HLGM generates higher SIB values across the images (with a few exceptions, as expected). Table III presents the median SIB values for the Oxford-IIIT dataset, spanning a range from 0.5 to 10. This table highlights that HLGM consistently achieves a higher median in the SIB distribution, signifying its superior performance.

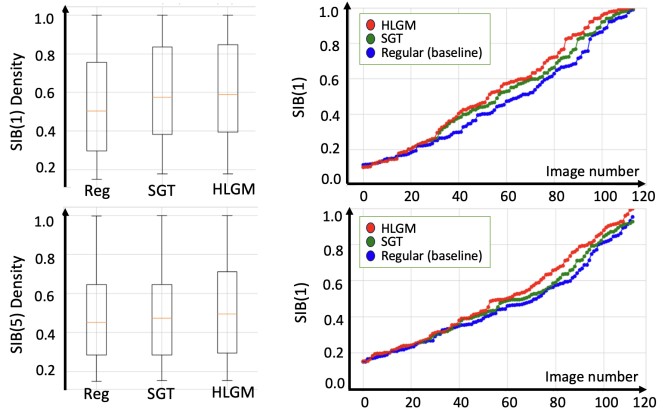

Fig. 8: (Left) Plot comparing SIB(1) and SIB(5) metrics across regular (baseline), SGT, and HLGM training methods, showing a notably higher mean for HLGM. (Right) Scatterplot of SIB values for 120 randomly chosen test images, illustrating the superior performance of HLGM in achieving higher SIB values compared to regular and SGT training methods.

## VI. CONCLUSION

In our study, we present HLGM, a novel saliency-guided training method that simultaneously boosts a model's accuracy and the quality of its saliency maps. This approach distinguishes itself from previous saliency-guided methods, which typically sacrifice accuracy in comparison to conventional training methods. HLGM, on the other hand, not only maintains but enhances model accuracy and is particularly adept at creating more refined saliency maps. To substantiate our assertions, we devised a unique metric to assess the capability of models in producing high-fidelity saliency feature maps that align precisely with the objects marked in the images. Employing this metric, we compared HLGM with standard baseline models and contemporary leading techniques. The findings show marked improvements with HLGM, underscoring its effectiveness. Overall, HLGM represents a significant step forward in training methodologies. It not only strives to outperform traditional training methods in accuracy but also contributes to greater model transparency through enhanced saliency mapping. This combination makes HLGM an influential asset in the field of machine learning.

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
