# OpenReview forum: "HLGM: A Novel Methodology For Improving Model Accuracy Using Saliency-Guided High and Low Gradient Masking"
_IEEE.org/ICIST/2024/Conference — IEEE ICIST 2024 Conference Submission_

### Official Review · Reviewer_Ab6a · 2024-08-23
**All in all good, but there are still some problems.**

**Rating:** 8
**Confidence:** 4

**Review:**

The manuscript introduces a novel methodology called High and Low Gradient Masking (HLGM), a saliency-guided training approach designed to enhance both model accuracy and the quality of saliency maps in computer vision models. Unlike traditional methods that often compromise accuracy, HLGM employs a two-phase process: a regular training phase and an accuracy-boosting phase that alternates between masking high gradient information to encourage diverse learning pathways and masking low gradient information to reduce background noise. The paper validates the effectiveness of HLGM through comparative analysis, demonstrating significant improvements over baseline models and other advanced techniques in terms of both model accuracy and saliency mapping. The approach is tested across multiple datasets and model architectures, showing substantial improvements in model robustness and interpretability.

Comments:

1.Clearly differentiate HLGM from existing methods like SmoothGrad to better highlight its unique contributions.

2.Include statistical significance, such as confidence intervals, to strengthen the performance claims.

3.Simplify complex algorithm sections and consider adding diagrams for better understanding.

4.Discuss HLGM's limitations and propose future research directions to enhance its relevance.

---

### Official Review · Reviewer_np86 · 2024-08-23
**This paper proposed a novel saliency-guided training method that simultaneously boosts a model’s accuracy and the quality of its saliency maps.The obtained result is valuable and can be accepted if the following problems can be clarified.**

**Rating:** 7
**Confidence:** 5

**Review:**

(1) The main contributions and significance should be rewritten to illustrate the main ideas and main work of this paper.
(2) What's the difference between the 'Introduction' and the 'Related Works' ?
(3) Try to maintain the work flow of the paper, especially during transition between sections and subsections.
(4)The quality of language needs significant improvement, and professional editing may be necessary.

---

### Official Review · Reviewer_wcMN · 2024-08-24
**Review Comments for Manuscript No. 182**

**Rating:** 8
**Confidence:** 3

**Review:**

1. The manuscript mentions the High and Low Gradient Masking (HLGM) approach. Is this approach being proposed for the first time? The challenges arising from combining HLGM with the method in [12] need to be explained in more detail to better highlight the contributions of this work.

2. In pursuing higher model accuracy, how does the method avoid overfitting to specific details, which could compromise the quality and generalizability of the saliency maps? Has this trade-off been explored and quantified in your experiments?

3. In the EXPERIMENTS section, what exactly is referred to as the "regular model"? The authors should provide a clear explanation.

4. In TABLE II, why do some baseline methods exhibit higher model accuracy compared to SGT?

5. While the overall formatting of the manuscript is relatively well-done, the authors should ensure that the text size and style within the figures are consistent.

---

### Decision · Program_Chairs · 2024-09-08

Accept (Oral)